# Interlinking Bristol Based Models to Build Resilience to Climate Change

**John Stevens [1],\*, Rob Henderson [2], James Webber [3], Barry Evans [3], Albert Chen [3], Slobodan Djordjević [3], Daniel Sánchez-Muñoz [4] and José Domínguez-García [4]**

[1] Bristol City Council, Strategic City Transport, Flood Risk Management Team, 100 Temple Street, Bristol BS3 9FS, UK

[2] Wessex Water, Engineering & Construction, Claverton Down Rd, Bath BA2 7WW, UK; rob.henderson@wessexwater.co.uk

[3] Centre for Water Systems, University of Exeter, Exeter EX4 4QF, UK; J.Webber2@exeter.ac.uk (J.W.); B.Evans@exeter.ac.uk (B.E.); A.S.Chen@exeter.ac.uk (A.C.); S.Djordjevic@exeter.ac.uk (S.D.)

[4] IREC, Power Systems Department, Jardins de les Dones de Negre, 1, 2ª pl., Sant Adrià de Besòs, 08930 Barcelona, Spain; dsanchezm@irec.cat (D.S.-M.); jldominguez@irec.cat (J.D.-G.)

\* Correspondence: john.stevens@bristol.gov.uk

**Abstract:** Expanding populations and increased urbanisation are causing a strain on cities worldwide as they become more frequently and more severely affected by extreme weather conditions. Critical services and infrastructure are feeling increasing pressure to be maintained in a sustainable way under these combined stresses. Methods to better cope with these demanding factors are greatly needed now, and with the predicted impacts of climate change, further adaptation will become essential for the future. All cities comprise a complex of interdependent systems representing critical operations that cannot function properly independently, or be fully understood in isolation of one another. The consequences of localised flooding can become much more widespread due to the inter-relation of these connected systems. Due to reliance upon one another and this connectedness, an all-encompassing assessment is appropriate. Different model representations are available for different services and integrating these enables consideration of these cascading effects. In the case study city of Bristol, 1D and 2D hydraulic modelling predicting the location and severity of flooding has been used in conjunction with modelling of road traffic and energy supply by linking models established for these respective sectors. This enables identification of key vulnerabilities to prioritise resources and enhance city resilience against future sea-level rise and the more intense rainfall conditions anticipated.

**Keywords:** fluvial; pluvial; tidal; sewer; flood; risk; climate change; modelling; cascading effects

## 1. Introduction

The challenge facing cities includes the need for urban expansion to accommodate rising populations. This puts a heightened demand on existing infrastructure and this is particularly apparent in many older cities that were not designed for the modern-day population and climate, making them unable to cope with such pressures [1,2]. The effects of climate change are likely to make this impact much worse in the future [3]. More severe storms and prolonged wet or dry periods all increase the risk and likelihood of pluvial flooding problems being encountered [4–7]. Drier landscapes from longer periods of drought in summer will produce surfaces more susceptible to rapid run-off and the increased storms and likelihood of thunderstorms exacerbates this risk [8]. Warmer and wetter winters mean more prolonged periods of wet weather, raised groundwater tables and higher river

baseflows [9]. Rising sea-levels and heightened river flows are anticipated to create a more substantial threat from tidal and fluvial flooding sources, especially given the increased storm surge component associated with this [9].

In response to this, as well as many other well-known implications associated with climate change, Bristol City Council (BCC, Bristol, UK) has declared a climate emergency and has issued a Climate Emergency Action Plan as well as the Bristol Resilience Strategy to try and counteract and reduce these factors where possible [10,11]. Change in the climate is inevitable and is already being experienced to some extent; evaluating ways to adapt to this change is therefore essential. Work conducted on the EU RESilience to cope with Climate Change in Urban arEas (RESCCUE) [12] project with BCC and other key partners such as Wessex Water, the University of Exeter and IREC (Catalonia Institute for Energy Research, Barcelona, Spain) has made efforts to devise ways of assessing and managing increased flood-related climate risk and these will be elaborated upon below.

This article responds to increasing hazards by evaluating interdependencies in critical infrastructure and services functioning in the city of Bristol. In particular, the work focuses on the key elements of the existing drainage infrastructure, electricity supply system and road network. Roads also represent a significant conveyance mechanism for urban surface water (Fewtrell et al., 2011) [13]. During intense rainfall, these are likely to act as channels for exceedance from the sewer network. Similarly, the energy distribution network can be disrupted during flooding, leading to cascading damages and service interruption across many sectors of a city. Previous research has typically evaluated these systems independently (Pyatkova et al., 2018), however, it is apparent that safe and effective management of cities requires full consideration of interdependencies between complex and highly connected urban systems [14]. The aim of the work is to identify where the main vulnerabilities lie in areas of central Bristol and its immediate surrounds that are more prone to flooding through interlinked modelling, in order to develop adaptation plans to counteract this risk. The paper is structured through initially setting the case study city background and weather-related climatic threats it is faced with now and that which are anticipated in the future. It then goes on to define how these risks have been modelled and assessed and interprets the findings based on implications posed from the various sources of flooding to certain city services and specific areas of the city. Adaptation measures proposed to counteract this risk and attempt to relieve some of the effects of these impacts are then given further consideration.

*Bristol Case Study*

Bristol is located in the South of England, UK within the Severn River Basin District (see Figure 1) and is particularly vulnerable to tidal/fluvial flood risk, being subjected to the second-highest tidal range in the world from the Severn Estuary which influences the tidal River Avon as well as having significant surface water flood risks [15]. The River Avon shown in Figure 2 passes through Bristol from East to West, with a portion of the flow entering the Floating Harbour in the central area, which has a regulated water level and has complex interactions between incoming tides and river flows. The majority of river flow is diverted along the River Avon New Cut where it continues westwards and discharges into the Severn Estuary at Avonmouth. The "New Cut" is a man-made channel of the River Avon and was constructed in the early 19th century to allow the creation of the Floating Harbour which provides a permanent dock facility isolated from extreme tidal effects. Many tributaries of the Avon within Bristol are tidally influenced near their outfalls.

The city is also rapidly expanding; in recent years it has seen the second-largest rate of population growth in the UK, outside of London. Urban expansion and the threat from intense downpours, which are expected to become more frequent and of greater severity in the future, combined with sea-level rise will impact on critical drainage infrastructure and land drainage functions in Bristol. Improving urban resilience in the city can be achieved by the capability to anticipate, prepare for, respond to, and recover from these significant multi-hazard threats with minimum damage.

In order to achieve the above aims, adaptation plans (on all scales from strategic, operational to community-based including societal and economic impacts) have all been duly considered. The

underlying objective was to find ways in which to adapt to this shift in weather patterns and account for what is "the new norm" through a range of proactive and reactive responses. Analysis of the impact that high tides combined with heavy rainfall have through direct flood damage on riverside areas adjacent to the tidal River Avon (and its tributaries) fulfilled part of this assessment. As well as direct impacts, indirect impacts (on the operation of urban drainage systems for instance) were evaluated. This included analysis of flooding issues linked to tidally influenced sections of the sewer network, for example. Wider impacts, in respect of the cascading effects on critical city services, were also considered as a follow-on consequence of flooding. The way in which this was quantified and an overview of some of the outcomes is described in the following sections.

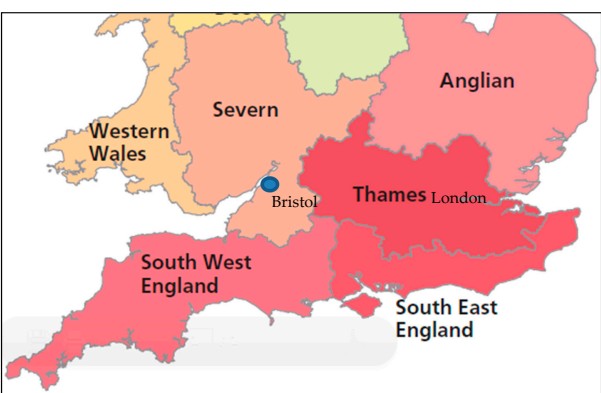

**Figure 1.** Location of Bristol, in the Severn River Basin District, shown on the South England and South Wales, UK River Basin District map.

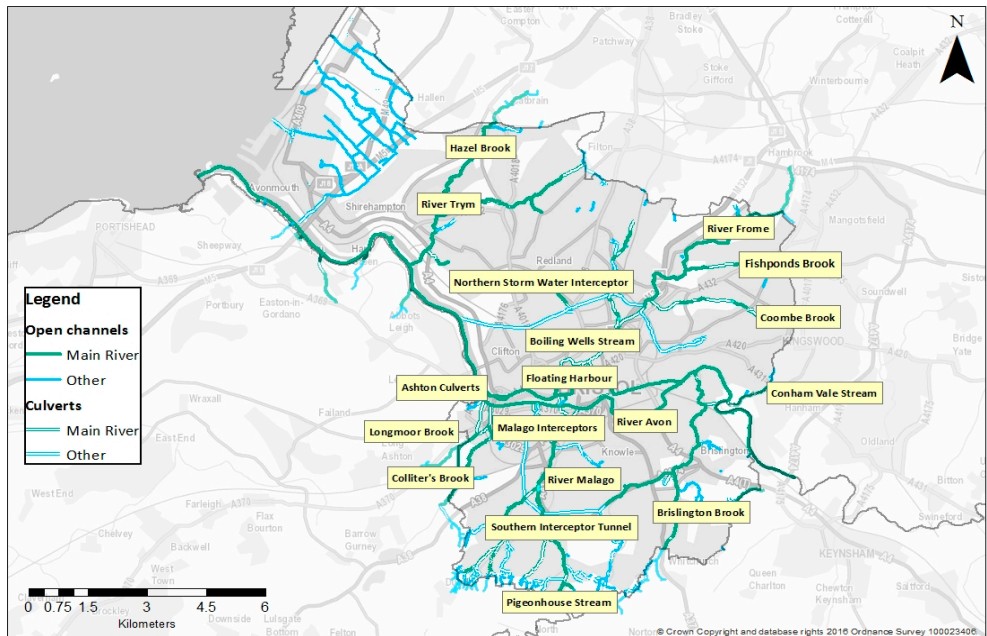

**Figure 2.** Map of main rivers, streams and surface water interceptor tunnels in the Bristol City Council Local Authority administrative area.

## 2. Materials and Methods

### 2.1. Flood Modelling

The city of Bristol is now quite comprehensively modelled as far as sewers, watercourses and the tidal River Avon is concerned. The Bristol Surface Water Management Plan [16] model, developed

in conjunction with Wessex Water, covers the entire city and incorporates much of the underground piped sewer network. The tidal River Avon is modelled throughout its expanse within the BCC area providing coverage of the whole watercourse within the city as modelled in the Central Area Flood Risk Assessment (CAFRA) [17]. Tributaries of the River Avon also have detailed flood mapping in the lower reaches of their catchments in the CAFRA [17]. At Avonmouth, in West Bristol, the effect of tidal and fluvial flooding from the Severn Estuary and Avonmouth rhyne network (drainage ditches that serve the area) is mapped through the Avonmouth/Severnside Level 2 Strategic Flood Risk Assessment [18]. Purely fluvial flood extents for other watercourses, not tidally influenced, that appear throughout the remainder of the city are covered by Environment Agency Flood Mapping [19]. By analysing the exposure of urban services and critical city infrastructure to flooding and the vulnerability of key services such as the electricity supply, road network and drainage infrastructure, the impacts and risk can be assessed over time and the necessary adaptation measures considered to ensure their continued functioning. The flood models for pluvial flooding as well as combined fluvial and tidal flood events that exist for Bristol can provide flood extents, levels, depths and hazard ratings inclusive of uplift for climate change. The models used of the sewerage systems for drainage aspects are coupled with models in the flooding sector, most notably with integrated modelling of drainage systems and watercourses. A hazard assessment for current and future (climate change) scenarios exists and this utilized detailed models and software tools including *Infoworks ICM* 1D/2D urban drainage modelling [20]. This analysis was based on models built during the preparation of the City Drainage Master Plans and Surface Water Management Plans. The tidal and fluvial model involved *ISIS* [21] and *TuFLOW* [22] for the joint probability modelling on the River Avon.

The estimated change in future climate parameters was informed by the Met Office and UK guidance and sensitivity checked by the Madrid-based Climate Research Foundation (FIC) as part of the RESCCUE project [12]. Models of the tidal and fluvial system in Bristol have been completed through the CAFRA [17] study conducted by BCC. CAFRA analyzed combination events of tidal floods and fluvial flood flows (i.e., the joint probability of the two flooding sources occurring simultaneously) of varying magnitudes. This was to establish the predominant risk and threat to the city centre, both now and into the future including the predicted impacts of climate change. The conclusion was that the high tidal element causes the greatest flooding risk, far outweighing the fluvial component. The CAFRA study included a large-scale hydraulic model of the tidal and fluvial systems in central Bristol. The model itself was completed using a combined 1D and 2D model built using *ISIS-TuFLOW* software packages. The majority of the river networks in central Bristol were simulated using the 1D *ISIS* software, with topography and ground surface represented using 2D *TuFLOW*. The model was updated as part of the ongoing River Avon Flood Risk Management Strategy.

The modelling has allowed flood depths, velocities and extents to be determined, allowing comprehensive flood hazard mapping for the city in accordance with the UK DeFRA standards [23,24]. Observed and predicted tidal flood levels for the Severn Estuary and tidal River Avon have provided some verification of the model outputs and a series of particularly high tides experienced in 2014 assisted with this, during a stage of the 19-year lunar cycle that caused exceptional astronomical tide levels. In order to predict and assess the likely impacts of climate change, the National Planning Policy Framework (NPPF) [25] and UKCP09 [26] derived uplifts have been applied to the CAFRA tidal/fluvial model. A damages assessment conducted in line with the "Multicoloured Handbook" (MCM) [27] methodology has allowed for the quantification of the flood damages incurred in monetary value in the present day and the future increase in damage value due to the effects of climate change. This involved land-use data acquired via the UK's National Receptor Database. The land-use includes MCM Codes that correspond to depth-damage curves for specified land-use types. Tangible damages on the economic sector were estimated by utilizing the SWMP pluvial model and CAFRA tidal/fluvial model compared against the land-use area distribution through the MCM approach.

Present-day Bristol faces a significant risk of flooding from multiple sources. With the application of a climate uplift factor applied to tide levels, fluvial river flows and rainfall intensity, this gives a

resulting increase in flood extents, depths, heightened flow velocities and subsequently an increase in flood hazard risk rating. The flood modelling for the present day has climate change allowances applied to it in line with UK Government guidance [28] to estimate the projected future flood risk. This was based on the data and recommendations available for the Severn River basin district to account for an anticipated increased peak river flow. Upper-end peak rainfall intensity increases were applied, applicable to all areas in England. For sea-level rise, the rate of increase (in mm per year–see Table 1) is reflected in accordance with the advice for the Severn River basin district to use the South West River basin allowances. The increases in flooding over time causes threats directly to land susceptible to this risk but the effects of this are also felt beyond immediate high flood-risk areas, as will be explained in this section. The quantification of damages, identification of key criticalities and vulnerabilities has highlighted the most vulnerable areas.

**Table 1.** Reflecting the predicted future sea-level rise from a *UK Government website* [28].

| Epoch | 1990 to 2025 | 2026 to 2055 | 2056 to 2085 | 2086 to 2115 | Cumulative Rise 1990 to 2115 (m) |
|---|---|---|---|---|---|
| Rate of rise (mm/yr) | 3.5 | 8 | 11.5 | 14.5 | 1.11 |
| Cumulative rise in epoch (compared to 1990) (mm) | 123 | 232 | 334 | 421 | |

The astonishing estimated increase in the tide levels over time is reflected in Table 1 which states national values recommended for planning purposes.

Table 2 defines the DeFRA/Environment Agency Flood Hazard rating and the danger posed to people, including the emergency services which could be called upon in times of flooding disruption caused to critical city services [23]. This has been used to assess the flood hazard posed in the Bristol case study areas.

**Table 2.** DeFRA/Environment Agency Flood Hazard rating and the danger posed to people for different combinations of flood depth and velocity [23].

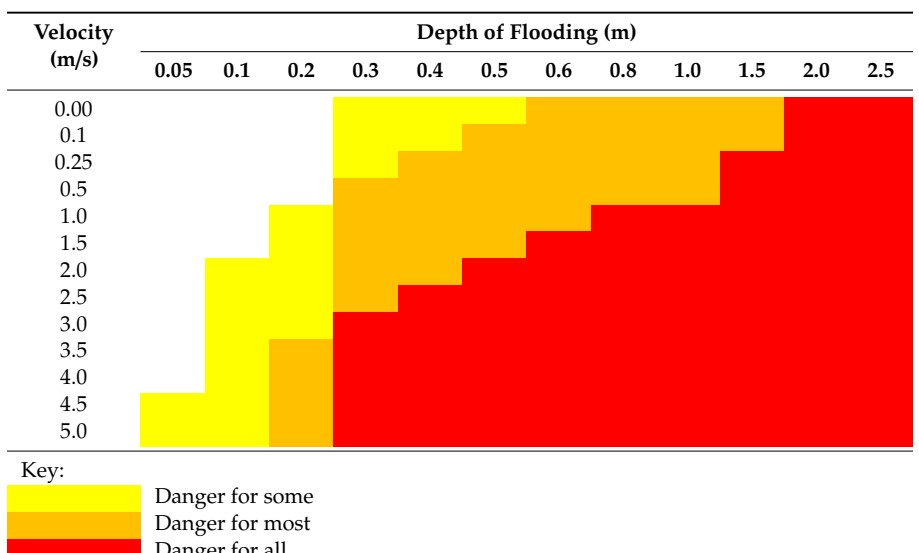

## 2.2. Integrated Modelling

As outlined in Figure 3, not only were the various sources of flooding combined but these were interfaced with models used to manage other city functions, such as traffic management and power supply by overlapping models that exist for these sectors, with the output of one model providing input to another. An integrated flooding-traffic model applied traffic simulations and flood impact

modelling carried out using the Simulation of Urban MObility (*SUMO)* [29] micro-scale traffic software package. The SUMO system can accommodate large road networks, appropriate for that modelled in Bristol, and provides a continuous traffic simulation using open source data [30–33].

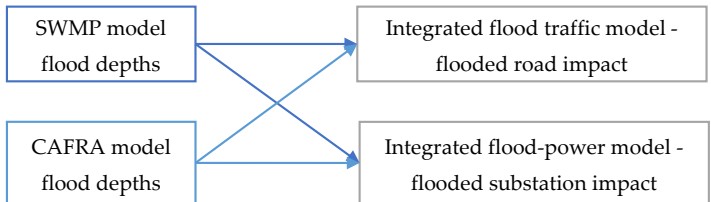

**Figure 3.** Schematic diagram showing the linkage between the integrated modelling.

Flooding impacts on traffic flow in several ways including: redirection of traffic, reduction in travel speeds, increases in journey times, congestion and increased pollution levels. The impacts may be localized or widespread as drivers try to avoid a problem area and, in doing so, cause congestion elsewhere [34,35].

The hazards posed to traffic flows are represented in relation to predicted flood depths along individual road segments/links in the *SUMO* model. Flood depths define whether a link (section of road) is closed (severe flooding) or if the maximum allowable speed along the link is to be reduced (moderate flooding). This shows which roads would be closed and which would suffer congestion and reduced speed. Any link that experiences flood depths of 0 to 0.10 m is determined as a non-affected road, 0.10–0.30 m is described as a reduced speed road link and road links with flooding of over 0.30 m are classified as closed. An indication of where the anticipated road closures would occur during a flood scenario can be inferred. In 100 years and with the potential effects of sea-level, the future effects of flooding can be surmised, allowing prediction of how the city will suffer in these areas in response to this. Network management plans can be devised in response to this in the current day or longer-term strategic solutions, and improved flood defences can be scoped out for the future.

The outputs of both the pluvial/sewer model and the tidal/fluvial models were also used to analyse the impact on the electricity supply system serving the central area of Bristol (8 km$^2$). The electrical modelling created a sampling layer through the use of the open-source software *QGIS* [36] using infrastructure location and attribute data provided by Western Power Distribution (WPD) [37], including critical flood-depth thresholds (where known) for individual stations. This integrated flooding-electrical model (*IFEM*), allows the generation of a GIS layer showing fully-affected, partially-affected and non-affected substations and their areas of influence [38]. Knowledge of flooding extents and depths allow the impact on urban services to be assessed in detail, thereby informing the planning of remedial and mitigation measures contributing to the development of a Resilience Action Plan (RAP).

By combining flood mapping and electrical modelling of the power network (using data derived from WPD), complications and cascading effects can be predicted. As the tidal cycle and future astronomical tide levels are forecast well in advance, high spring tides that may combine with adverse prevailing weather conditions can be foreseen with greater warning time ahead of a preceding tidal flood event. Low atmospheric pressure systems and westerly winds raise the tidal storm surge component in Bristol. Knowledge of these factors can then help in tidal flood preparations and electricity substations within the potential flood area can be identified and actions taken to mitigate or eliminate the flooding risk. Sewer flood maps and tidal/fluvial flood maps highlight how many substations could potentially be affected with an increased magnitude of flood events if there is no protection around the substations. The greater vulnerabilities and particular areas of concern can then be demonstrated from this and used to inform the selection of effective protection measures [39]. Impacts from the electricity supply system resulting in power outages further afield can be yet another implication and cascading effect felt by other city services reliant upon this facility.

Two particular high-risk areas within Bristol were then focused on to provide an in-depth detailed analysis at significant locations. The problematic areas were analysed to formulate a RAP to cope better with this and to enhance future sustainability. The impact of flooding on urban services helped quantify more of the overall risk faced. The two areas where analysis of traffic and energy disruption caused by flooding has been conducted are (a) St Phillips Marsh and (b) Ashton.

### 2.3. Focus Areas

St Phillips Marsh, located east of central Bristol tidal/ fluvial flooding is the principal risk to this mainly commercial and light industrial area. This area is heavily trafficked since it feeds in and out of the main central business hub of the city and is already inundated with commuters.

Ashton, located in South-West Bristol where flooding occurs from the main watercourse (Colliters Brook) and sewer systems (both combined and surface water); flooding is also significantly affected by the water level in the tidal River Avon due to backing up of drainage systems in this low-lying part of the city. Detailed *Infoworks* ICM-2D modelling was undertaken within this subcatchment.

## 3. Results

### 3.1. Sources of Flooding Modelled

#### 3.1.1. Tidal Fluvial Flooding

In the present day, there are currently 1000 properties shown as "at risk" to an extreme (1 in 200-year Return Period (RP) or 0.5% Annual Exceedance Probability (AEP)) tidal event in Bristol. The future number of properties at risk rises to 3700 in the eventuality of an extreme (1 in 200-year RP or 0.5% AEP) tidal event in Bristol becoming apparent in the future (2115) when considering the predicted effects of sea-level rise.

The increase in predicted flood extents over the coming decades is illustrated in Figures 4–6. The flow of the River Avon through central Bristol is from East to West, discharging in the Severn Estuary. The tidal influences already cause flooding from the River Avon and Bristol Floating Harbour during exceptional high spring tides and sea-level rise will exacerbate this problem in the future.

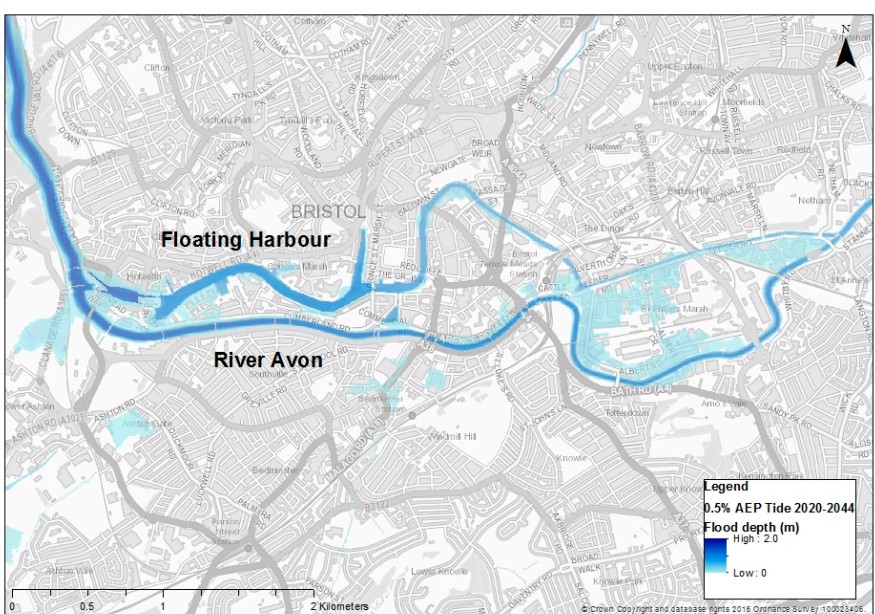

**Figure 4.** Flood depths and extent for a 0.5% AEP tidal flood event for the 2010–2044 scenario.

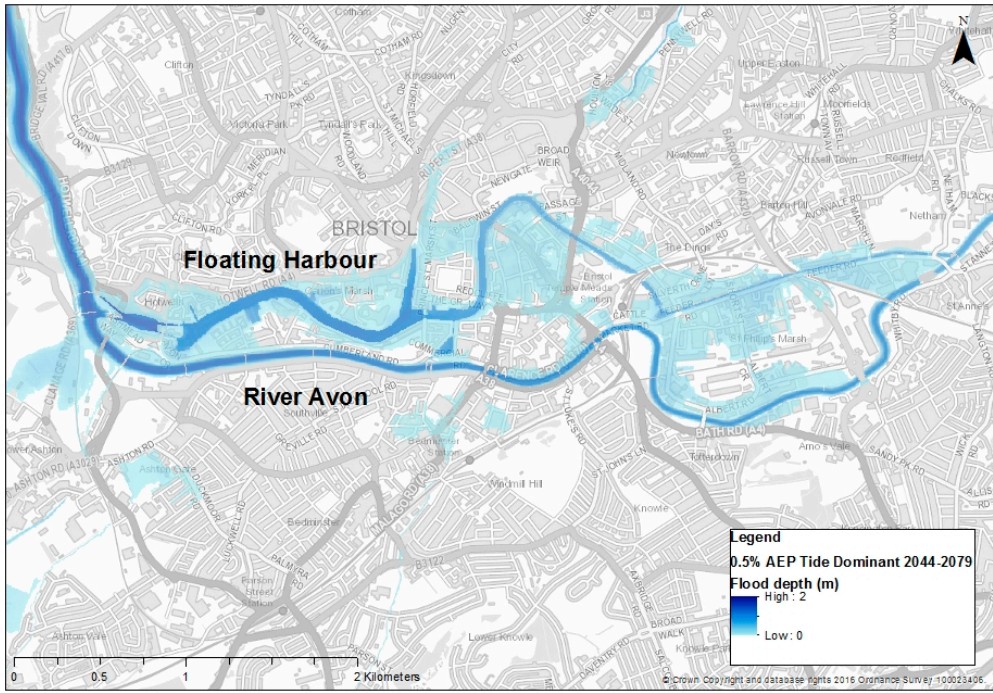

**Figure 5.** Flood depths and extent for a 0.5% AEP tidal flood event for the 2044–2079 scenario.

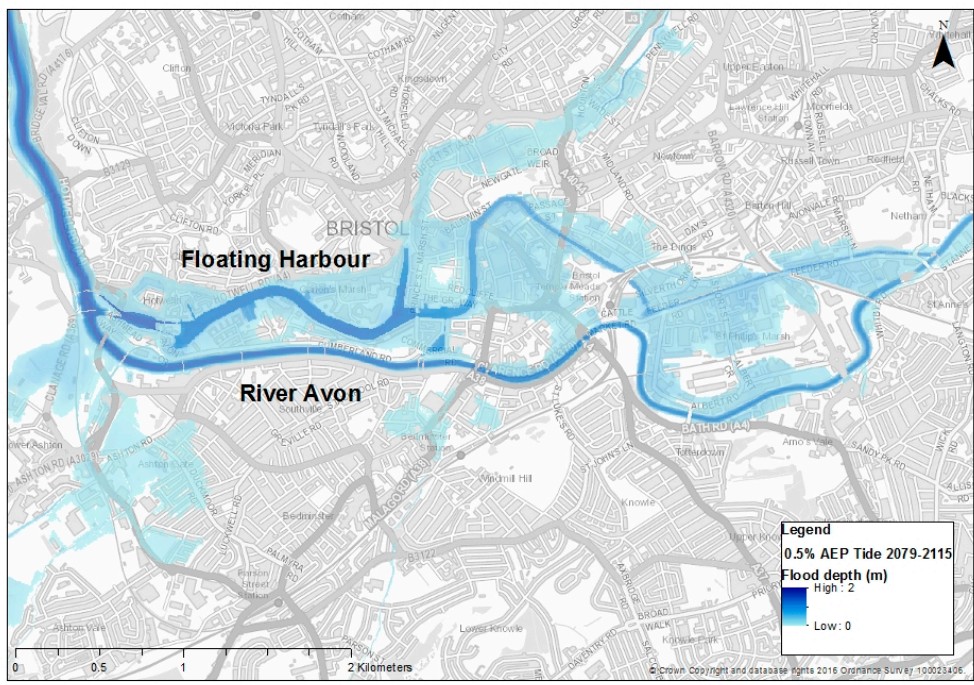

**Figure 6.** Flood depths and extent for a 0.5% AEP tidal flood event for the 2079–2115 scenario.

In respect of flood hazard mapping, it is evident that the potential impact of climate change (primarily sea-level rise) will have huge implications for properties at risk and for the continuity of city services. Figure 7a,b displays this.

The DeFRA methodology for assessing flood hazard is accepted for application in the UK, applicable to Bristol. Flood hazards posing a danger to people have also been assessed, however, through other means in the works conducted by Martinez-Gomariz E et al. (2016), Chanson H and Brown R (2015), Russo B et al. (2013), Arrighi C et al. (2017) that could be more applicable at other localities [40–43].

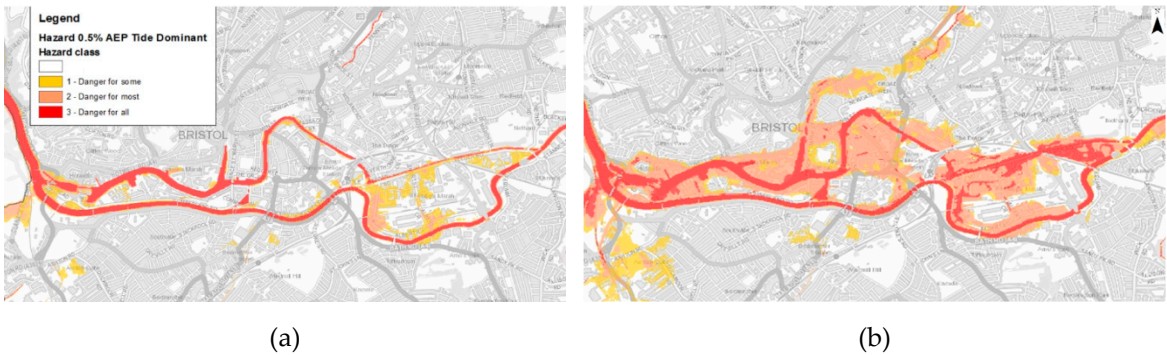

(a)                                                                    (b)

**Figure 7.** Hazard mapping for the 0.5% AEP tidal/fluvial event in (**a**) Present day (left) (**b**) 2115 (inclusive of climate change) (right).

### 3.1.2. Pluvial Flooding

More intense storms in the magnitude of 20–40% greater intensity are expected to occur more commonly by the turn of the next century [28]. This element is captured in the increased flood extents illustrated in Figure 8. In general, these show that under climate change and for very rare events, the depth and severity of flooding will increase more significantly than will the extents of flooding, due to the constraints imposed by the urban terrain.

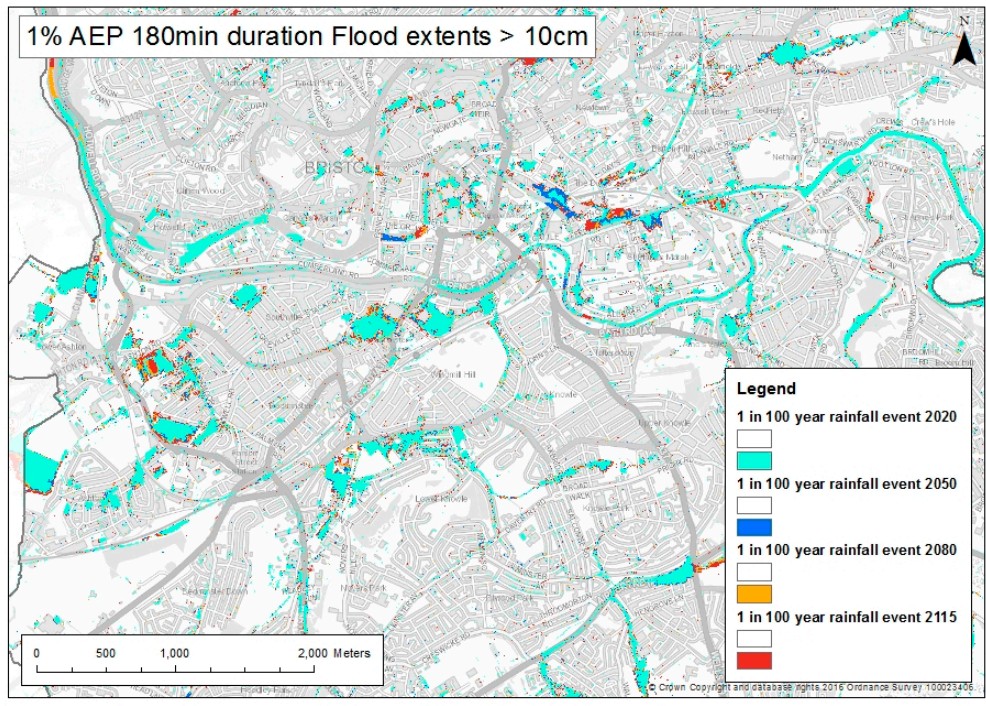

**Figure 8.** Pluvial flood extents increasing over time with the predicted impacts of climate change.

In the Ashton area, the likelihood of property flooding as a result of combining a high tide with a severe storm is estimated to be roughly four times more probable than at present by the 2050s and over ten times more probable by the 2080s under the climate change conditions assessed here.

### 3.2. Integrated Flood Models

Flood models are useful inputs for evaluating the potential impact on critical city services such as traffic and power supply. This then helps address the most vital elements and assists in targeting limited

resources more effectively. It also helps promote the business case to encourage longer-term investment in strategic interventions or to help devise shorter-term remedies such as operational procedures.

### 3.2.1. Traffic

For pluvial flooding, the analysis in the Ashton area showed that for a 1 in 30-year rainfall event, (3.3% AEP) journey times went up considerably with the time of flood duration causing more prolonged traffic disruption. Delay times on the network increased by approximately 14, 24 and 25 min for the 20, 30 and 40 min duration rainfall events, respectively.

For fluvial flooding, the increase in reduced speed journeys and closed roads are shown in Figures 9 and 10 below.

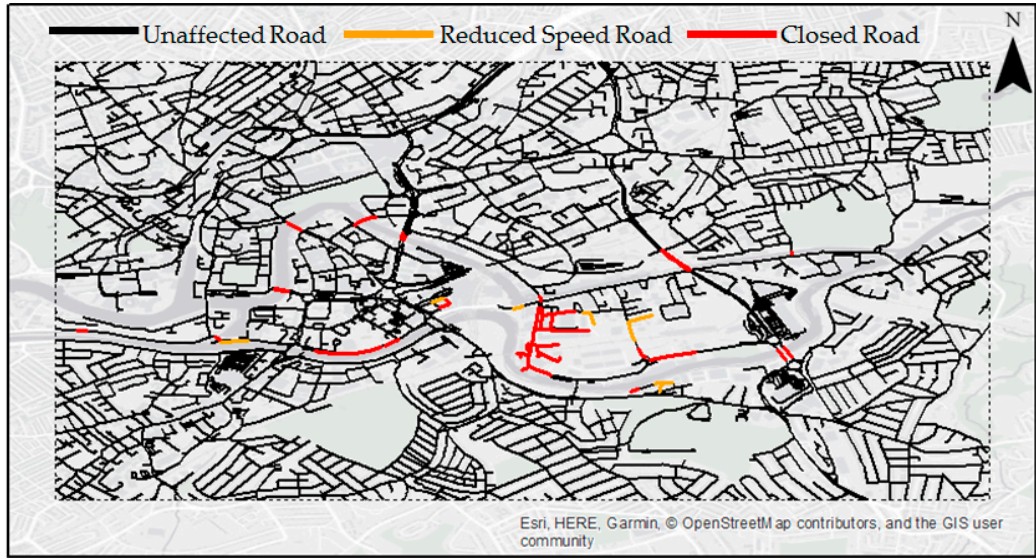

**Figure 9.** Flooding on the St Philip's Marsh Bristol Road Network for a 1 in 20 Year Fluvial Current Day event day.

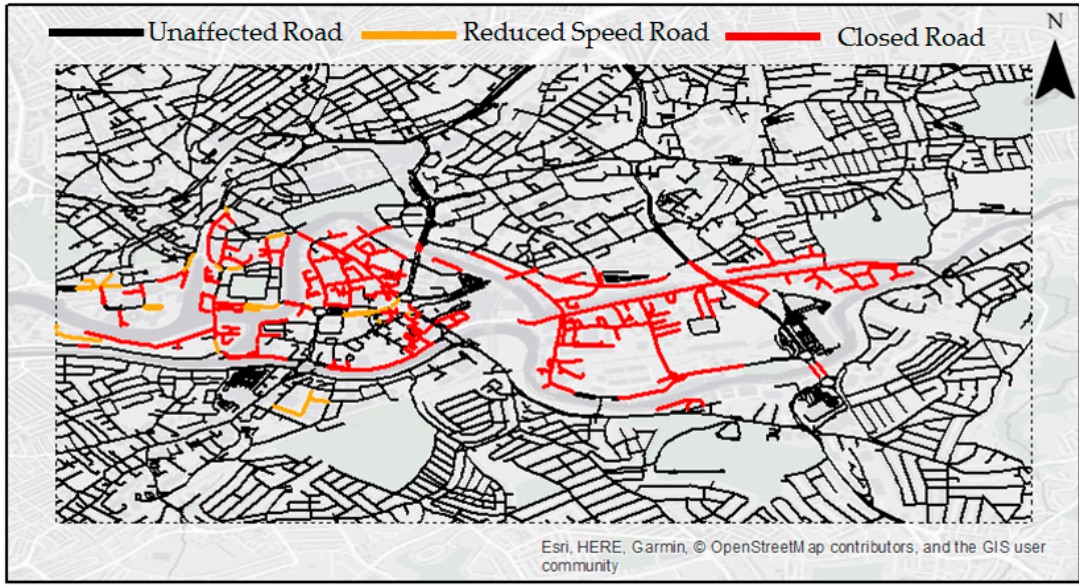

**Figure 10.** Flooding on the St Philip's Marsh Bristol Road Network for a 1 in 20 Year Fluvial Future Climate Change Event 2115.

### 3.2.2. Power

Utilising the location of critical infrastructure (such as electricity substations) and comparing against predicted flood outlines can help infer power outages associated with flood events. Substations located within the CAFRA and SWMP flood model outlines within central Bristol, St Philips and Ashton were analysed. Specific information relating to the criticality of individual major substations cannot be disclosed in greater detail due to security reasons, but the number of potential substations affected and the percentage over the total studied are given in Table 3 for the three Average Water Depth (AWD) categories established. In the most critical case (AWD > 1.60 m), the increase in severe flooding occurrences rises from 2 to 76 when increasing the return period from T20 to T1000, meaning a 17.2% increase over the total number of 11 kV substations. These results are also displayed in Figure 11.

**Table 3.** Number of substations affected resulting from the electrical sector analysis, and the percentage over the total of substations studied, according to different return periods, type of substations and average water depth (AWD) categories.

| Water Depth | Type of Substation | Number of Occurrences | | | Percentage over Total | | |
|---|---|---|---|---|---|---|---|
| | | *T20* | *T200* | *T1000* | *T20* | *T20* | *T200* |
| 0.1 m < AWD ≤ 0.8 m | *11 kV* | 80 | 49 | 41 | 18.6% | 11.4% | 9.5% |
| | *33 kV* | 0 | 1 | 1 | 0.0% | 33.3% | 33.3% |
| | *132 kV* | 0 | 0 | 0 | 0.0% | 0.0% | 0.0% |
| 0.8 m < AWD ≤ 1.6 m | *11 kV* | 19 | 92 | 87 | 4.4% | 21.4% | 20.2% |
| | *132 kV* | 0 | 1 | 1 | 0.0% | 100.0% | 100.0% |
| AWD > 1.6 m | *11kV* | 2 | 56 | 76 | 0.5% | 13.0% | 17.7% |

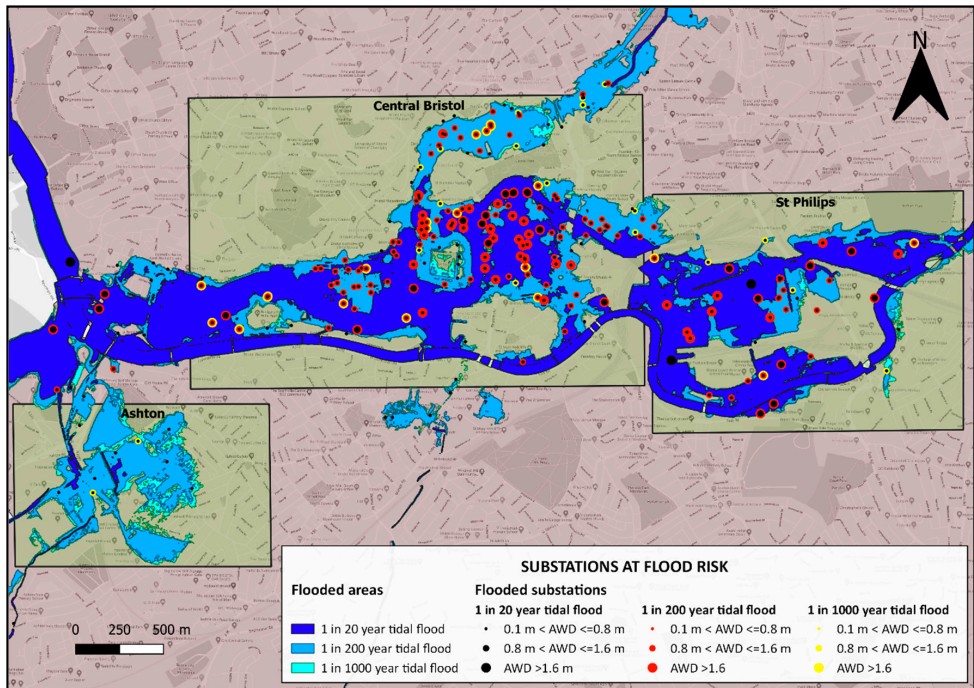

**Figure 11.** Map of substations potentially affected by flooding for various return periods in the 2115 future case scenarios. The points are sized according to the three categories of flood event magnitude established and coloured according to the year. This map shows the evolution of the flooding depending on the return period.

### 3.3. Focus Areas

#### 3.3.1. St Philips Marsh

The benefits of protecting the riverside low spots subsequently by removing the over-spilling flood extents shown in Figure 12 in the Bristol central area were quantified through a damages assessment. Present-day damages are estimated through the MCM in the order of £40M whereas in 100 years with the rate of sea-level rise continuing, this will be around £400M. The cost-benefit ratio can be gauged from this when scoping out flood defences that may be suitable, feasible and economically affordable at this locality.

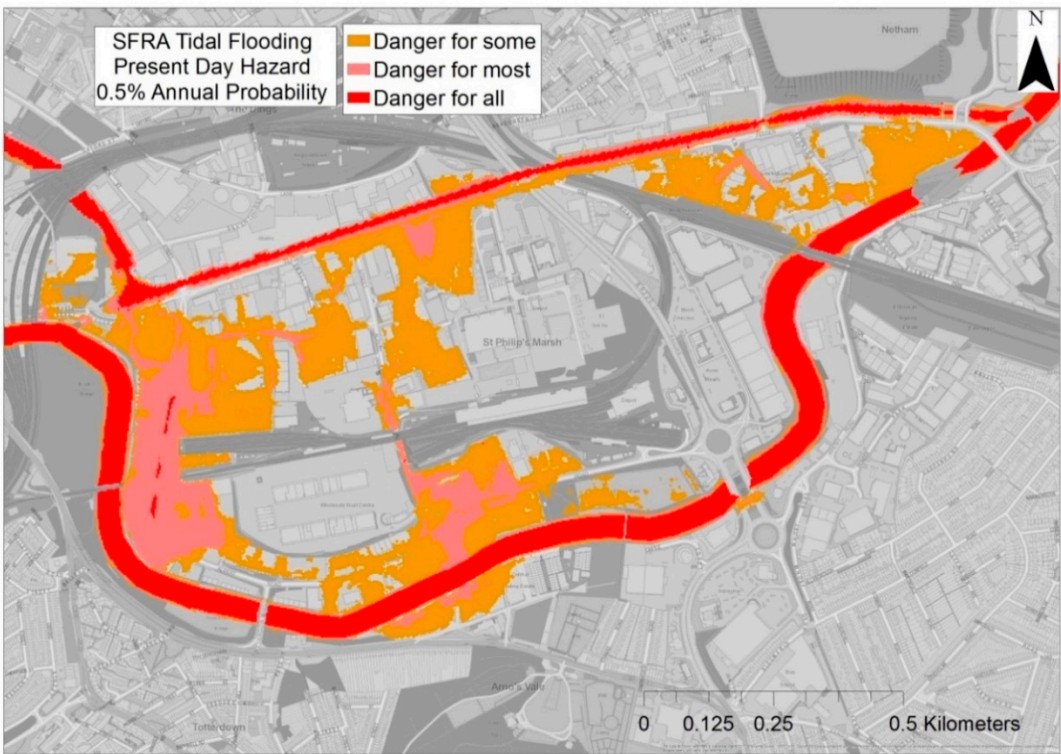

**Figure 12.** DeFRA/Environment Agency Flood Hazard Mapping showing the danger posed to people (please refer to Table 2) at St Philips Marsh for a 0.5%AEP tidal flood event.

#### 3.3.2. Ashton

The increased pluvial flood depths and extents are very noticeable in the Ashton area in Figure 13 and the future tidal/fluvial flood hazard shown in Figure 14.

Critical parts of the Ashton area are at a lower elevation than the banks of the River Avon. The River Avon New Cut river channel banks are at an elevation of between 8.5 m and 14.0 m AOD. By contrast, the lowest ground elevation in Ashton is 6.3–6.8 m AOD (public open space/parkland), with roads and properties at 6.8–7.5 m AOD or higher. The river level frequently surpasses this level in the present-day during high Spring tides. In the extreme scenario, a (current) 1:200 year tidal/fluvial flood event could take the river level to about 9 m AOD, well over two metres higher than the lowest vulnerable ground level and would inundate the neighbourhood. With the effects of sea-level rise in 100 years, another metre may be added to the extreme tidal flood level. The average duration for which these critical levels are exceeded is reflected numerically in Table 4 and then visually in Figure 15 to outline the level of flooding experienced.

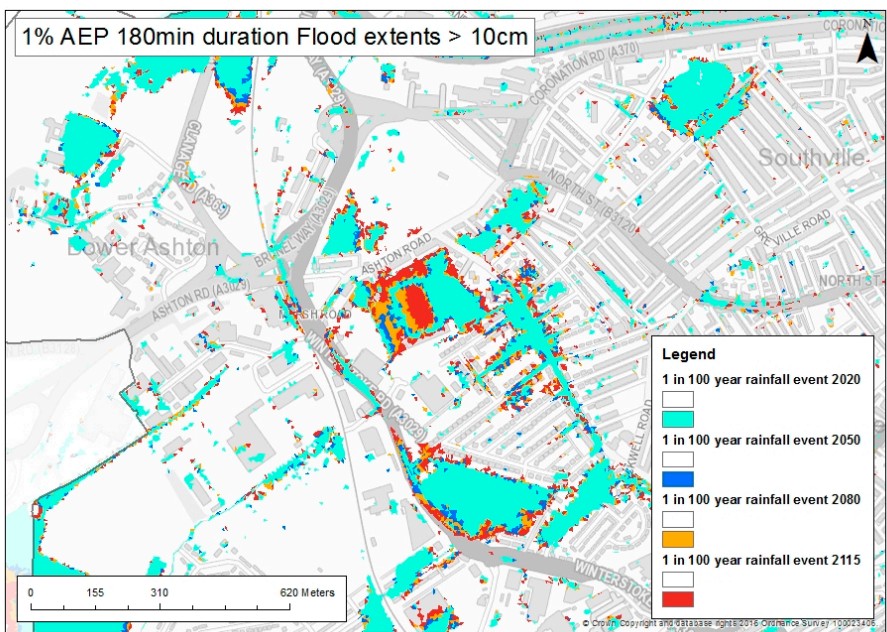

**Figure 13.** Flooding extents in the Ashton area with climate uplifts (Note: this model run excluded tidal effects in order to identify changes due to rainfall increase alone).

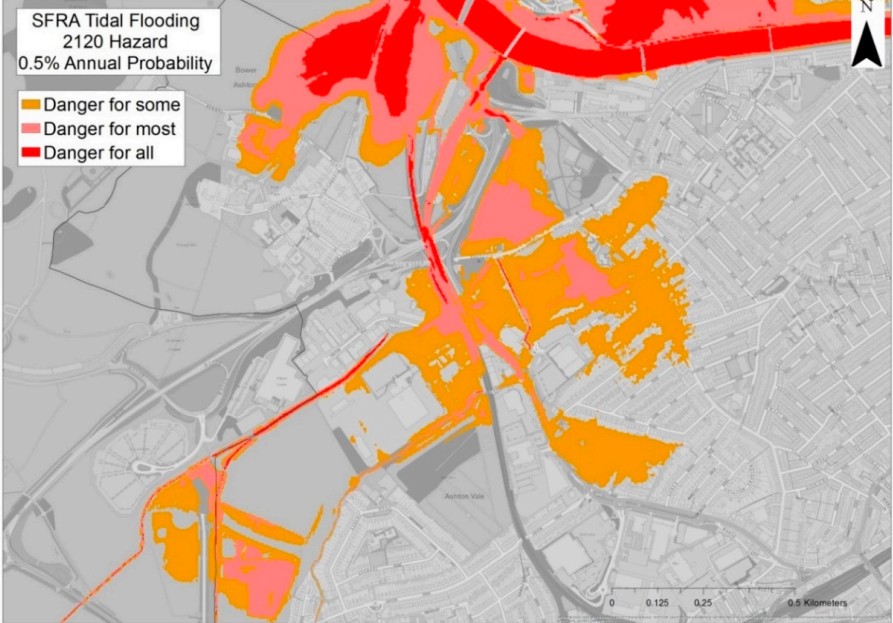

**Figure 14.** DeFRA/Environment Agency Flood Hazard Mapping showing the danger posed to people (please refer to Table 2) at Ashton for a 0.5% AEP tidal flood event in 2120 inclusive of climate change.

**Table 4.** Critical tide durations estimated in future epochs.

| Decade | Critical Tide Level (7.5mAOD) Is Exceeded (%, Percentage of Time in a Year, on Average) | Extreme Tide Level (8.0mAOD) Is Exceeded (%, Percentage of Time in a Year, on Average) |
|---|---|---|
| 2010 | 0.34 | 0.04 |
| 2050 | 0.63 | 0.13 |
| 2080 | 1.30 | 0.39 |
| 2110 | 2.04 | 0.79 |

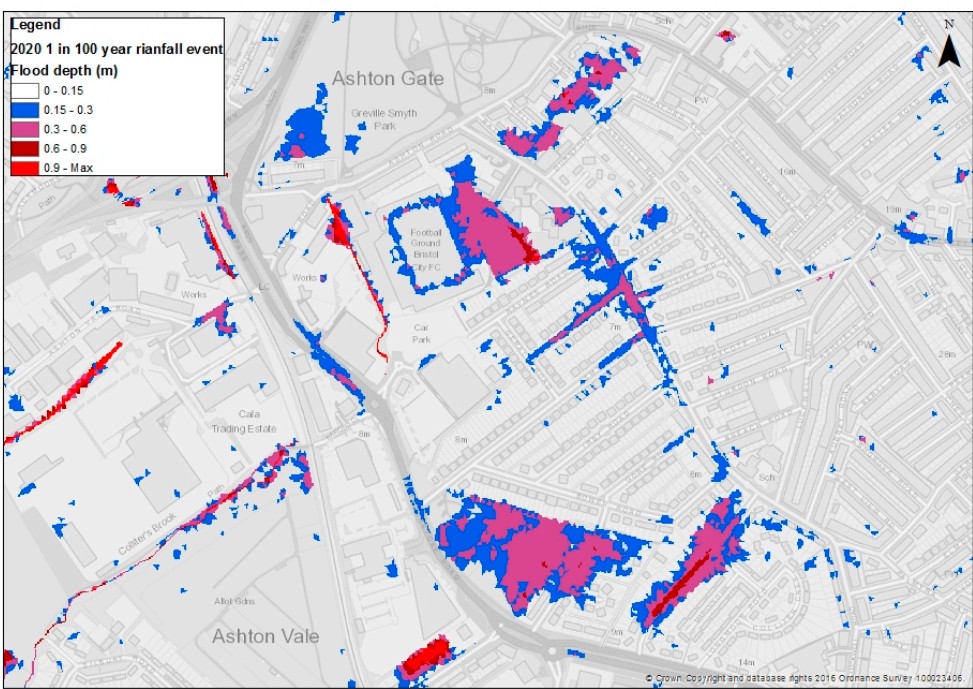

**Figure 15.** Flooding in the Ashton area in the current day.

The complex interactions between drainage systems at Ashton include the influences of the river tide level on surface water and combined sewer overflow (CSO) outfalls causing "tidelocking" (the closure of non-return valves), resulting in backing up and surcharging of the system. Overflow from a culverted watercourse to the combined sewer system is another contributory factor as is the discharge from CSOs that will naturally increase with heightened rainfall. Inflow from natural watercourses to man-made drainage ditches, surface water sewers and culverts also occurs as does flooding out of watercourse channels to urban surfaces and flooding out of (combined and surface water) sewers to urban surfaces. The 2D modelling of these systems has improved the understanding of the complex interactions between surface flows and the drainage systems.

The Colliters Brook and surface water sewers discharge to the River Avon by gravity outfalls, protected by tide flaps. When high tide level exceeds the outfall level, flows back up within the Colliters Brook (which is in a culvert upstream of the outfall). Similarly, flows back up within the surface water systems discharging to the River Avon or the Colliters Brook. With increasing sea levels, surface water systems with tidal river outfalls will be compromised under high tide conditions, Figures 15 and 16 demonstrate this.

A major sewage pumping station (SPS) at Ashton Avenue takes combined sewage flow from the Ashton area which helps alleviate the existing local flood risk issues. During intense storms, if the pumping capacity is exceeded, flows are diverted to a gravity overflow system installed with tide flaps. The gravity overflow can only discharge when the energy head in the surcharged trunk sewer is higher than the river level. In extreme conditions (when all pumping capacity is beaten and very high tide conditions prevail), flow level in the sewer can back up-potentially to ground surface level. Thus, flooding of low-lying areas from the combined system could occur if the storm is of sufficient intensity/duration and is coincident with a high river level. This increases the risk of combined sewer flooding in severe storms which exceed the pumping capacity at Ashton Avenue SPS. Total pumping capacity is currently exceeded by a storm of roughly 1 in 5 years or greater under current rainfall conditions, with significant flooding predicted to occur roughly once in 30 years. Diversion of "clean" streamwater to the combined system also increases CSO spill frequency at the SPS under lesser storm events.

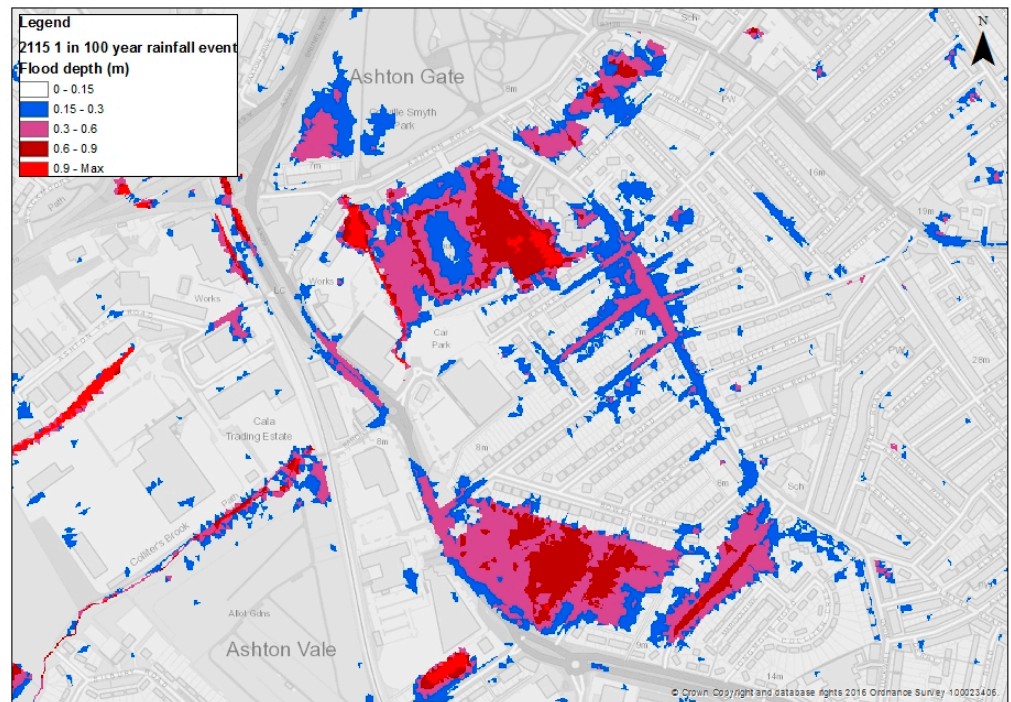

**Figure 16.** Flooding in the Ashton area for the 2115 scenario (inclusive of climate change).

Climate change poses two direct threats to flooding in Ashton:

- With projected sea-level rise, the duration of critical tidal levels (exceeding about 7.5 m AOD) will be longer, thus the gravity overflow at Ashton Avenue SPS will be able to operate for less time (as indicated in Table 4)
- Severe storms and wet-weather periods will be more frequent and intense, increasing the likelihood of
- Sewer flows exceeding the installed sewage pumping capacity
- Slightly higher flood flow levels in the river on top of the tidal effects

Within 100 years, the duration of a tide which is likely to cause flooding when the Ashton Avenue pumping station is beaten (i.e., 7.5 m AOD) shows a 6 x increase in the probability of occurrence. Furthermore, the duration of the tide which could cause serious flooding of properties (i.e., the 8.0 m AOD tide) would increase from 0.04% to 0.79% of the time–a 20× increase in the likelihood of occurrence compared to present day.

Sewer modelling has also indicated that under expected future rainfall conditions, the total installed pumping capacity at the Ashton Avenue pumping station could be exceeded roughly as follows:

- Current = exceeded once in 5 years
- 2050s = exceeded once in 2–3 years
- 2080s = exceeded once in 1–2 years
- 2110s = exceeded about once per year

## 4. Discussion

The main outputs of the SWMP model, that is, larger areas of pluvial flooding, have been verified by observations on-site during heavy rainfall events. The re-runs of the 2018 version of the SWMP model were also compared to the 2012 edition and the two correlated well. Through this analysis and interpretation of the results, an outlined package of adaptation measures and strategies based on these findings has been formulated. Examples of adaptations included in Ashton are:

- Provision of a surface water pumping facility to allow the watercourse/surface water system to discharge at all states of the tide (including sea-level rise)
- Re-grading of the Colliter's Brook open channel section to alter the gradient and widen the channel, providing more conveyance capacity
- Reinstatement of syphons and modifications to the structure which currently allows overflow from the watercourse/surface water system to the combined system
- Reducing the impermeable area in subcatchment upstream of Ashton Drive by 20% achievable through the introduction of sustainable drainage systems

  Central area and St Philips Marsh:

- Construction of riverside flood defence walls to protect the low spots

Many of the outputs, methods and principles could be applied to any disruptive threats to the normal running of a city, thus allowing improved capacity to respond to shock events. Trying to reduce these impacts or enhance the recovery time by gaining a greater understanding of these systems and connections will offer improvement. Projections of key climate variables (rainfall, temperature) and sea-level rise for the epochs generated and how they will highlight the fragility and limitations on existing infrastructure and service functions were demonstrated. Knowledge of the problems faced assists in developing ways to sustain our key city functions and operations. Benefits of conducting the analysis include highlighting the criticality of points of the transport network for network management plans in the current day, such as the redirecting of traffic and issuing road closures that can be enacted ahead of a high tide warning. Another benefit is in identifying flooded electricity substations which, when resulting in failure, could indeed impact on another service that is reliant upon it as well within the wider network that it serves, and which may be well outside of the original flooded area. To make use of this, further investigation is required into other connected services. A key finding of the analysis is that there is a need for extra sewer and/or land-drainage pumping station pumping capacity to serve the area of Ashton in order to cope with future climate and tidal conditions. This is in addition to other means of reducing flood flows entering the area such as separation and use of SuDS.

The modelling begins to demonstrate the complexities of the city once different overlying functions are considered together as one. Understanding the 'domino effect' can then begin to quantify the cascading implications. Evaluating these connections can help build resilience in developing emergency response procedures and additionally inform strategic interventions. By integrating the models of urban management systems with flood models, an overall projection of city risk can be portrayed. Gaining a greater understanding of the resilience of city systems when we encounter disruptive events like flooding can, in principle, be applied in a similar process to other physical, social and economic challenges too. They will experience disturbances under such flood scenarios in the current day, which will worsen further still in the future with the predicted effects of climate change. From this, we can try and predict what some of the impacts will be if we were to experience extreme flood events, assess this and make plans to try and counteract it.

## 5. Conclusions

The key findings of this study include the need for an essential improvement of existing drainage infrastructure serving the area of Ashton in Bristol. The predicted effects of climate change and in particular, the impact of sea-level rise on tidal outfalls, will mean that a critical pumping station operating in the area will provide diminishing protection against extreme events as time progresses. The current pumping capacity will fail to deal effectively with the more intense storms and heightened river flows anticipated in future when outfalls become increasingly strained under rising sea-levels.

In the central St Philip's Marsh area, the "dry island" effect posed to this locality will have a larger knock-on impact to wider traffic flows in the adjoining road networks. In the future, under high spring tide or extreme tidal flood conditions, road closures will be far more prevalent and journey time

delays escalated. Network management procedures will need to adapt to this and the requirement for longer-term strategic flood defences will become more pressing.

The number of electricity substations in central Bristol has shown to be at increased vulnerability to extreme tidal flooding when future flood extents and depths are considered highlighting how in the worst case (AWD > 160 cm) the number of substations affected could increase from 2 to 76 when increasing the return period from T20 to T1000. The substations identified as within the at-risk zones will need to ensure localized flood protection of the substation units up to the predicted future flood levels.

From this paper, it can be seen the importance and need for an integrated analysis for risk assessment related to city management during extreme events. Additionally, such analysis is of extreme relevance in order to detect the most critical zones and elements which may be impacted during such events; allowing to define and develop corrective strategies within the city with a holistic view.

**Author Contributions:** J.S. and R.H.; writing—original draft preparation, formal analysis, B.E., A.C., S.D., J.W., D.S.-M. and J.D.-G.; formal analysis, writing—review and editing. All authors have read and agreed to the published version of the manuscript.

**Funding:** This work was framed into the RESCCUE project, funded by the European Union's Horizon 2020 Research and Innovation Programme under Grant Agreement Number 700174.

**Acknowledgments:** The authors would like to acknowledge the RESCCUE project partner organisations and service providers operating in the city of Bristol, Western Power Distribution.

**Conflicts of Interest:** The authors declare no conflict of interest.

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
