# Peer review of "Interlinking Bristol Based Models to Build Resilience to Climate Change"

_sustainability, doi:10.3390/su12083233_

Round 1
Reviewer 1 Report
The paper presents a case study of urban resilience in the context of climate change, so it is of interest of the journal. As it is, the reviewer does not think the manuscript is ready for publication, but resubmission is encouraged.
Main concerns include:
- The paper is not detailed, it looks rushed and not refined, the structure is weak and figures are very approximate. The academic standard is low.
- The paper lacks literature review: the Introduction has not references, and there is no section dedicated to review existing studies, motivation or aim.
- The specific aim is unclear: is this the analysis of the vulnerability of the electricity supply and road system in case of flooding?
Further comments:
L4: Some authors are missing
L32 and after: references are needed; cardinal references (North, East, etc.) need capital letters
L81-90: can authors report some evidence of this through papers/reports/links?
L113: need reference and more details for SUMO
L93: please give a reference for the project
L140: authors should detail better the use of the uplifting factors. How were these obtained?
Figure 2: the quality is very poor, there is no legend. A navigator to show Bristol in the UK would be useful to an international audience. The a/b/c indication is not present, although used in the main text (e.g. Fig. 2a)
Figure 3: danger for “some/most/all” is not clear. Authors should quantify the risk more clearly. The figure is not crossreferenced in the main text.
Figure 4: legend has not clear labels.
Figure 5: poor figure, legend labels unclear, no geographical reference.
L220: how can “complications and cascading effect” be predicted?
Figure 6: legend labels are unclear
Figures 7, 9 and 10: poor quality, no legend
L237: the MCM Book should be explained in the Methods
L311-312: please give examples of adaptation measures.
L321-326: this study was not needed to state these two points
L327-331: the study did not demonstrate these two points
Whole manuscript: please check double spaces.
Reviewer 2 Report
Please see the attached file

Round 2
Reviewer 1 Report
I thank the authors for this revision that much improved the paper. However, I don’t think the paper is ready for publication yet. The paper is very “rich”: a very clear structure, very clear definitions, very snappy and summarised sections are needed in order to avoid confusion and flash out the paper importance. Honestly, I think there is perhaps too much material for a single paper (16 figures, including a b c etc!) and I would suggest authors to think about that in order to not “badly sell” their research, which is actually very valuable. It is clear that figures have been done by different people and/or software, and some of them have really poor quality.
As the paper is now, the science is not reproducible due to the lack of details.
L42-117: I don’t understand what is going here
Fig. 1: need “Bristol” close to the blue dot, and I would show also London. Which type of region are shown? (e.g. county, basin, etc)
Fig. 2: show Bristol and key locations
L201: specify the aim again
L213/272: use sub-sections, and avoid to begin paragraphs with “Word word: [text]”
Sec 2: input and output are still not well-explained, e.g. via flowchart. I would do that in Fig. 3: would perhaps this explained the connection between the models?
L328-334: make clear for which area the electric network is analysed
Fig 5: please (even slightly) separate pics a and b
Table 2: please specify in the captions (and in the text) if the hazard rating is for people or vehicles or … - and how this was used in the study.
L464-466: how the flood depths have been defined? How did authors determine road closure? In other words, please define “non affected roads”, “reduced speed roads” and “closed roads”
Fig 7 (caption): using a different palette colour would help to convey the meaning of the pics. For example, try to use black/grey for non affected roads, orange for reduced speed road and red for closed road I would try to integrate the caption in both Fig 7a and 7b
L519: it should be Average Water Depth, shouldn’t be?
L520-21: do you have a reference for this method?
Fig 8: would it be possible to have a zoom on St Philips Marsh or Asthon as well?
Fig 9: does this refer to Table 2? It is not clear nor explained. Danger for who/what? Please specify in the caption the represented scenario.
Fig 11: please improve the caption.
Sec 5: could authors quantify results with “numbers”, e.g. number of power station, minutes of delay, close roads, etc?
All figures: could authors uniform the symbol of the North and in general try to get a similar graphics?
Reviewer 2 Report
The authors have resolved the issues. However, the responses were mixed with the main article. The authors should separate them before submitting the manuscript for publication.
Round 3
Reviewer 1 Report
I appreciate the efforts put in this new version, however I am afraid (because I know the pain of being reviewed three times and still rejected!) that the standard is still far from a scientific publication. The paper is rushed, superficial and approximate in many aspects; the structure is complicated with multiple (not useful) sub-headings. The quality of figures is low. Moreover, I have realised there is confusion between the concepts of “hazard” and “risk” (produce of hazard, vulnerability and exposure).
I am listing below further suggestions, but I recommend to review the paper deeply, asking perhaps to colleagues to provide independent revision too.
Fig. 1: the dpi of the figure is low (and the label “Bristol” far from its location). The dot is missing at the end of the caption.
Fig. 2: I think the text of yellow labels is too small.
Fig 3: still unclear inputs/outputs.
L195: I never seen “:” followed by a new sub-section. I would do: (a) and (b). Specify if both (energy and traffic) analysis are carried out for both locations.
L209-221: not sure this is needed here.
L222-224: there are three level of “headings” here, 3.1 > 3.1.1 > 1. Please simplify (also L269 and afterwards)
L223: the sub-sections is “risk”, however Fig. 4 (a/b/c) show the hazard only.
Table 1: has this been developed by this piece of research? If not, it should not stay in “Results”
Figure 4: this is divided into a), b) and c). As such, there should be a unique figure (which includes the three pics, that need to be labelled in a corner with a/b/c) and one caption (“Figure 4) that explains ..a), … b) …. and c). Authors should comment more about the difference between Fig. 4a, 4b and 4c.
L249-268: for the “danger to people”, there is much more literature available that should at least be mentioned. See e.g.: Martinez-Gomariz E et al (2016) “Experimental study of the stability of pedestrians exposed to urban pluvial flooding work; Chanson H and Brown R (2015) “New criterion for the stability of a human body in floodwaters”; Russo B et al (2013) “Pedestrian hazard criteria for flooded urban areas”; Arrighi C et al (2017) “Hydrodynamics of pedestrians' instability in floodwaters”.
Table 2: this is not part of the results, it should be introduced in the literature and said to be used in the methodology. The results should focus on the outcome from the application of it.
L269: again, the sub-section is about “risk” and the content is about hazard (i.e. flood extents)
L284: flood models “produce” useful inputs? It is not clear how/if the maps show above are integrated into the traffic and power model
L290-305: the method behind SUMO should go into the methodology, not in the results. Moreover, the description of how impact is assessed (i.e. flood depth impact roads, LL298-300) is very subjective; there is literature available (e.g. Pregnolato et al. 2017 “The impact of flooding on road transport: A depth-disruption function).
L318: more measures (or clearer) of disruption are needed. Which journeys do the 14/24/25 min of delay relate to? How many roads are closed (on the total)? Etc
L322-345: again, the methodology should go into the method section
Table 3: this is good, and something similar should be done for traffic
Fig. 7: please us the same names for the location as in the main text (i.e. add Vale and Marsh)
L364: I don’t understand what this section is about. Integrate it into the results above or specify better why it is needed to stay separated.
Fig. 9: is this Fig developed by DEFRA? If so, please remove from the results since it is not an outcome of the study.
L558-559: which results of the study demonstrated this? Please used obtained numbers.
L565-566: see comment above
Round 4
Reviewer 1 Report
I don't really how to say (again) that the structure is extra-complicated and the reading not smooth. Some comments are still ignored and scientific explanation not given. For me, there is too much material in this paper (16 figures, 16!) and authors could have written two papers with more clear structure and results.
Below,
Fig. 1: maybe it is a joke, but I meant a normal dot "." at the end of the caption!
L206-210: still - on which basis 0-10cm as assumed as open roads, and 10-30cm are assumed as reduced speed road? how is this reduced speed obtained? Techniques and methods needs to be scientific and explained, as authors did for danger to people.
L249-253: are two sub-sections really needed for 9 lines?
L260-319 (and later): why are outputs for flood modelling different, namely "hazard mapping", "flood depths" and "flood extents" (e.g. fig. 6 and 7 and 8)? Do we need the sub-sub-subsection up to the 4th level?! (e.g. 3.1.1.2)
L322-324: this should be integrated into the literature and explain why the DEFRA method is preferred instead of these others
L365-367: still - what does "considerably" mean? for which journeys the "delay times" have been "increased by approximately 14, 24 and 25 367 minutes for the 20, 30 and 40 minute duration rainfall events, respectively"?
L369-370: how many roads were closed/speed-reduced for the different scenarios? Indeed I suggested to do something similar to Table 3. Not sure what the answer "That is not possible at this time" does mean.
